# Time Matters in Regularizing Deep Networks:
## Weight Decay and Data Augmentation Affect Early Learning Dynamics, Matter Little Near Convergence

**Aditya Golatkar, Alessandro Achille, Stefano Soatto**
Department of Computer Science
University of California, Los Angeles
{aditya29,achille,soatto}@cs.ucla.edu

## Abstract

Regularization is typically understood as improving generalization by altering the landscape of local extrema to which the model eventually converges. Deep neural networks (DNNs), however, challenge this view: We show that removing regularization after an initial transient period has little effect on generalization, even if the final loss landscape is the same as if there had been no regularization. In some cases, generalization even improves after interrupting regularization. Conversely, if regularization is applied only after the initial transient, it has no effect on the final solution, whose generalization gap is as bad as if regularization never happened. This suggests that what matters for training deep networks is not just whether or how, but *when* to regularize. The phenomena we observe are manifest in different datasets (CIFAR-10, CIFAR-100, SVHN, ImageNet), different architectures (ResNet-18, All-CNN), different regularization methods (weight decay, data augmentation, mixup), different learning rate schedules (exponential, piece-wise constant). They collectively suggest that there is a "critical period" for regularizing deep networks that is decisive of the final performance. More analysis should, therefore, focus on the transient rather than asymptotic behavior of learning.

## 1 Introduction

There is no shortage of literature on *what* regularizers to use when training deep neural networks and *how* they affect the loss landscape but, to the best of our knowledge, no work has addressed *when* to apply regularization. We test the hypothesis that applying regularization at different epochs of training can yield different outcomes. Our curiosity stems from recent observations suggesting that the early epochs of training are decisive of the outcome of learning with a deep neural network [1].

We find that regularization via weight decay or data augmentation has the same effect on generalization when applied *only* during the initial epochs of training. Conversely, if regularization is applied only in the latter phase of convergence, it has little effect on the final solution, whose generalization is as bad as if regularization never happened. This suggests that, contrary to classical models, the mechanism by which regularization affects generalization in deep networks is not by changing the landscape of critical points at convergence, but by influencing the early transient of learning. This is unlike convex optimization (linear regression, support vector machines) where the transient is irrelevant.

In short, what matters for training deep networks is not just whether or how, but *when* to regularize.

In particular, the effect of temporary regularization on the final performance is maximal during an initial "critical period." This mimics other phenomena affecting the learning process which, albeit temporary, can permanently affect the final outcome if applied at the right time, as observed in a variety of learning systems, from artificial deep neural networks to biological ones. We use the methodology of [1] to regress the most critical epochs for various architectures and datasets.

Specifically, our findings are:

  (i) Applying weight decay or data augmentation beyond the initial transient of training does not improve generalization (Figure 1, Left). The transient is decisive of asymptotic performance.

 (ii) Applying regularization only during the final phases of convergence does not improve, and in some cases degrades generalization. Hence, regularization in deep networks does not work by re-shaping the loss function at convergence (Figure 1, Center).

(iii) Applying regularization only during a short sliding window shows that its effect is most pronounced during a critical period of few epochs (Figure 1, Right). Hence, the analysis of regularization in Deep Learning should focus on the transient, rather than asymptotics.

The explanation for these phenomena is not as simple as the solution being stuck in some local minimum: When turning regularization on or off after the critical period, the value of the weights changes, so the solution moves in the loss landscape. However, test accuracy, hence generalization, does not change. Adding regularization after the critical period *does* change the loss function, and also changes the final solution, but not for the better. Thus, the role of regularization is not to bias the final solution towards critical points with better generalization. Instead, it is to bias the initial transient towards regions of the loss landscape that contains multiple equivalent solutions with good generalization properties.

In the next section we place our observations in the context of prior related work, then introduce some of the nomenclature and notation (Sect. 3) before describing our experiments in Sect. 4. We discuss the results in Sect. 5.

## 2  Related Work

There is a considerable volume of work addressing regularization in deep networks, too vast to review here. Most of the efforts are towards analyzing the geometry and topology of the loss landscape at convergence. Work relating the local curvature of the loss around the point of convergence to regularization ("flat minima" [14, 21, 8, 4]) has been especially influential [5, 24]. Other work addresses the topological characteristics of the point of convergence (minima vs. saddles [6]). [20, 16] discuss the effects of the learning rate and batch size on stochastic gradient descent (SGD) dynamics and generalization. At the other end of the spectrum, there is complementary work addressing initialization of deep networks, [10, 13]. There is limited work addressing the *timing* of regularization, other than for the scheduling of learning rates [32, 25].

Changing the regularizer during training is common practice in many fields, and can be done in a variety of ways, either pre-scheduled – as in homotopy continuation methods [28], or in a manner that depends on the state of learning – as in adaptive regularization [18]. For example, in variational stereo-view reconstruction, regularization of the reconstruction loss is typically varied during the optimization, starting with high regularization and, ideally, ending with no regularization. This is quite unlike the case of Deep Learning: Stereo is ill-posed, as the object of inference (the disparity field) is infinite-dimensional and not smooth due to occluding boundaries. So, ideally one would *not* want to impose regularization, except for wading through the myriad of local mimima due to local self-similarity in images. Imposing regularization all along, however, causes over-smoothing, whereas the ground-truth disparity field is typically discontinuous. So, regularization is introduced initially and then removed to capture fine details. In other words, the ideal loss is not regularized, and regularization is introduced artificially to improve transient performance. In the case of machine learning, regularization is often interpreted as a prior on the solution. Thus, regularization is part of the problem formulation, rather than the mechanics of its solution.

Also related to our work, there have been attempts to interpret the mechanisms of action of certain regularization methods, such as weight decay [38, 35, 26, 15, 23, 3], data augmentation [36], dropout [34]. It has been pointed out in [38] that the Gauss-Newton norm correlates with generalization, and with the Fisher Information Matrix [9, 2], a measure of the flatness of the minimum, to conclude that the Fisher Information at convergence correlates with generalization. However, there is no causal link proven. In fact, we suggest this correlation may be an epi-phenomenon: Weight decay causes an increase in Fisher information during the transient, which is responsible for generalization (Figure 5), whereas the asymptotic value of the Fisher norm (*i.e.*, sharpness of the minimum) is not causative. In particular, we show that increasing Fisher Information can actually improve generalization.

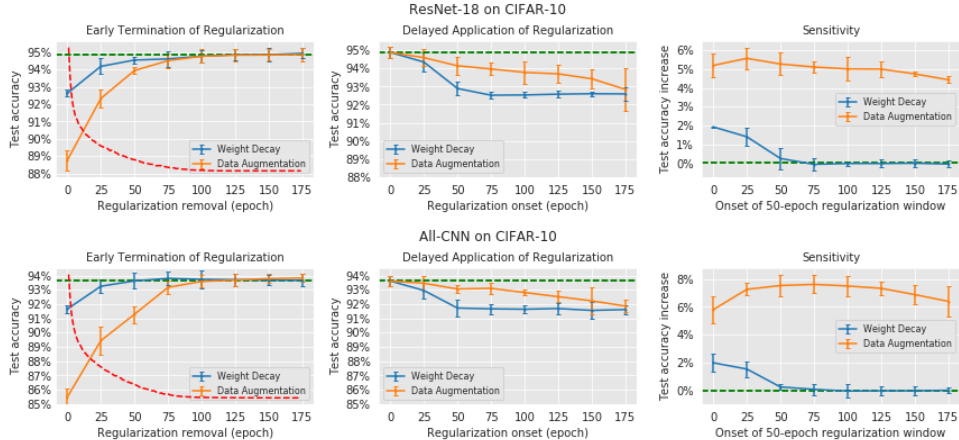

Figure 1: **Critical periods for regularization in DNNs :** **(Left)** Final test accuracy as a function of the epoch in which the regularizer is removed during training. Applying regularization beyond the initial transient of training (around 100 epochs) produces no appreciable increase in the test accuracy. In some cases, early removal of regularization *e.g.*, at epoch 75 for All-CNN, actually improves generalization. Despite the loss landscape at convergence being un-regularized, the network achieves accuracy comparable to a regularized one. **(Center)** Final test accuracy as a function of the onset of regularization. Applying regularization after the initial transient changes the convergence point (Fig. 2, B), but does not improve regularization. Thus, regularization does not influence generalization by re-shaping the loss landscape near the eventual solution. Instead, regularization biases the solution towards regions with good generalization properties during the initial transient. Weight decay (blue) shows a more marked time dependency than data augmentation (orange). The dashed line (green) in (Left) and (Center) corresponds to the final accuracy when we regularize throughout the training. **(Right)** Sensitivity (change in the final accuracy relative to un-regularized training) as a function of the onset of a 50-epoch regularization window. Initial learning epochs are more sensitive to weight decay compared to the intermediate training epochs for data augmentation. The shape of the sensitivity curve depends on the regularization scheme as well as the network architecture. For experiments with weight decay (or data augmentation), we apply data augmentation (or weight decay) throughout the training. Critical period for regularization occurs during the initial rapid decreasing phase of the training loss (red dotted line), which in this case is from epoch 0 to 75. The error bars indicate thrice the standard deviation across 5 independent trials.

## 3 Preliminaries and notation

Given an observed input $x$ (*e.g.*, an image) and a random variable $y$ we are trying to infer (*e.g.*, a discrete label), we denote with $p_w(y|x)$ the output distribution of a deep network parameterized by weights $w$. For discrete $y$, we usually have $p_w(y|x) = \text{softmax}(f_w(x))$ for some parametric function $f_w(x)$. Given a dataset $\mathcal{D} = \{(x_i, y_i)\}_{i=1}^{N}$, the cross-entropy loss of the network $p_w(y|x)$ on the dataset $\mathcal{D}$ is defined as $L_{\mathcal{D}}(w) := \frac{1}{N} \sum_{i=1}^{N} \ell(y, f_w(x_i)) = \mathbb{E}_{(x_i, y_i) \sim \mathcal{D}}[-\log p_w(y_i|x_i)]$.

When minimizing $L_{\mathcal{D}}(w)$ with stochastic gradient descent (SGD), we update the weights $w$ with an estimate of the gradient computed from a small number of samples (mini-batch). That is, $w_{t+1} \leftarrow w_t - \eta \mathbb{E}_{i \in \xi_t}[\nabla \ell(y_i, f_w(x_i))]$ where $\xi_t \subseteq \{1, \ldots, N\}$ is a random subset of indices of size $|\xi_t| = B$ (mini-batch size). In our implementation, weight decay (WD) is equivalent to imposing a penalty to the $L_2$ norm of the weights, so that we minimize the regularized loss $\mathcal{L} = L_{\mathcal{D}}(w) + \frac{\lambda}{2}\|w\|^2$.

Data augmentation (DA) expands the training set by choosing a set of random transformations of the data, $x' = g(x)$ (*e.g.*, random translations, rotations, reflections of the domain and affine transformations of the range of the images), sampled from a known distribution $P_g$, to yield $\mathcal{D}'(g) = \{(g_j(x_i), y_i)\}_{g_j \sim P_g}$.

In our experiments, we choose $g$ to be random cropping and horizontal flipping (reflections) of the images; $\mathcal{D}$ are the CIFAR-10 and CIFAR-100 datasets [22], and the class of functions $f_w$ are ResNet-18 [12] and All-CNN [33]. For all experiments, unless otherwise noted, we train with SGD

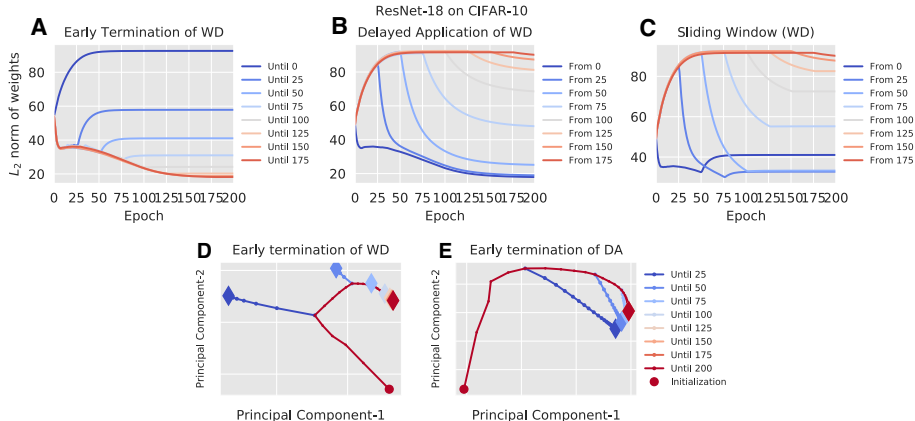

Figure 2: **Intermediate application or removal of regularization affects the final solution: (A-C)** $L_2$ norm of the weights as a function of the training epoch (corresponding to Figure 1 (Top)). The weights of the network move after application or removal of regularization, which can be seen by the change in their norm. Correlation between the norm of the weights and generalization properties is not as straightforward as lower norm implying better generalization. For instance, **(C)** applying weight decay only at the beginning (curve 0) reduces the norm only during the critical period, and yields higher norm asymptotically than, for example, curve 25. Yet it has better generalization. This suggests that the having a lower norm mostly help only during the critical period. We plot the norm of the weights for 200 training epochs to confirm that the weights stabilize and would not improve further with additional training. **(D)** PCA-projection of the training paths obtained removing weight decay at different times (see Appendix A.1). Removing WD *before* the end of the critical period (curves 25, 50) makes the network converge to different regions of the parameter space. Removing WD *after* the critical period (curves 75 to 200) still sensibly changes the final point (in particular, critical periods are not due the optimization being stuck in a local minimum), but all points lie in a similar area, supporting the Critical Period interpretation of [1]. **(E)** Same plots, but for DA, which unlike WD does not have a sharp critical period: all training paths converge to a similar area.

with momentum 0.9 and exponentially decaying learning rate with factor $\gamma = 0.97$ per epoch, starting from learning rate $\eta = 0.1$ (see also Appendix A).

## 4 Experiments

To test the hypothesis that regularization can have different effects when applied at different epochs of training, we perform three kinds of experiments. In the first, we apply regularization up to a certain point, and then switch off the regularizer. In the second, we initially forgo regularization, and switch it on only after a certain number of epochs. In the third, we apply regularization for a short window during the training process. We describe these three experiments in order, before discussing the effect of batch normalization, and analyzing changes in the loss landscape during training using local curvature (Fisher Information).

**Regularization interrupted.** We train standard DNN architectures (ResNet-18/All-CNN on CIFAR-10) using weight decay (WD) during the first $t_0$ epochs, then continue without WD. Similarly, we augment the dataset (DA) up to $t_0$ epochs, past which we revert to the original training set. We train both the architectures for 200 epochs. In all cases, the training loss converges to essentially zero for all values of $t_0$. We then examine the final test accuracy as a function of $t_0$ (Figure 1, Left). We observe that applying regularization beyond the initial transient (around 100 epochs) produces no measurable improvement in generalization (test accuracy). In Figure 3 (Left), we observe similar results for a different data distribution (CIFAR-100). Surprisingly, limiting regularization to the initial learning epochs yields final test accuracy that is as good as that achieved by regularizing to the end, even if the final loss landscapes, and hence the minima encountered at convergence, are different.

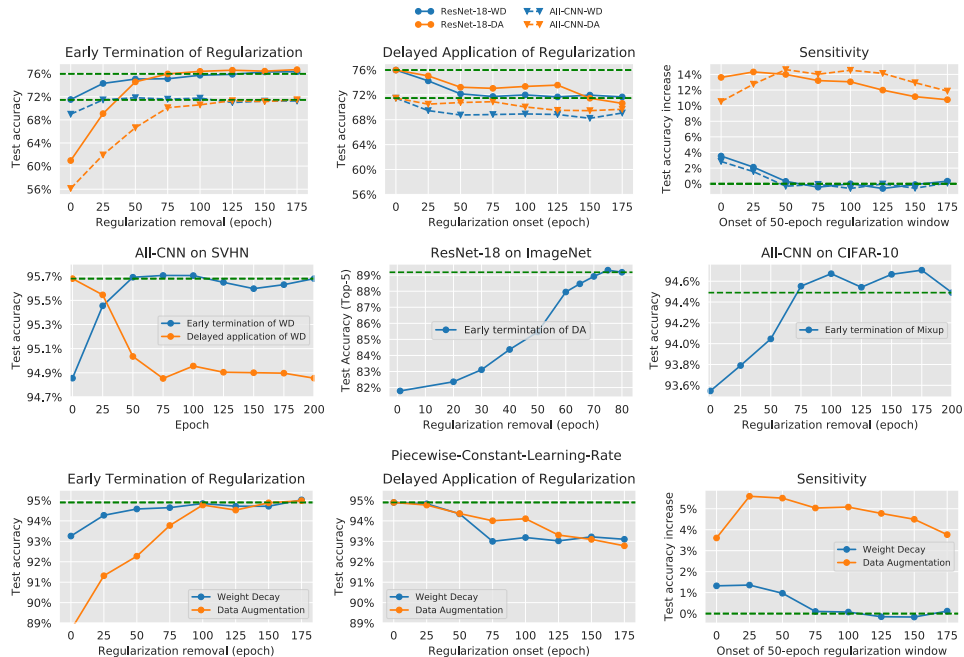

Figure 3: **(Top) Critical periods for regularization are independent of the data distribution:** We repeat the same experiment as in Figure 1 on CIFAR-100. We observe that the results are consistent with Figure 1. The dashed line (green) in (Left) and (Right) denotes the final accuracy when regularization is applied throughout the training. The dashed line on top corresponds to ResNet-18, while the one below it corresponds to All-CNN. **(Center)** In the middle row (Left and Center), we show critical regularization periods for models trained on SVHN [30] and ImageNet [7]. Critical periods for regularization also exists for regularization methods apart from weight decay and data augmentation, for example, Mixup [39] (Center Right). In fact, we observe that applying Mixup only during the critical period (first 75-100 epochs) results in better generalization compared to applying it throughout the training. **(Bottom) Critical regularization periods with a piecewise constant learning rate schedule:** We repeat experiment in Figure 1, but change the learning rate scheduling. Networks trained with piecewise constant learning rate exhibit behavior that is qualitatively similar to the exponentially decaying learning rate. The same experiment with constant learning rate is inconclusive since the network does not converge (see Appendix, Figure 11).

It is tempting to ascribe the imperviousness to regularization in the latter epochs of training (Figure 1, Left) to the optimization being stuck in a local minimum. After all, the decreased learning rate, or the shape of the loss around the minimum, could prevent the solution from moving. However, Figure 2 (A, curves 75/100) shows that the norm of the weights changes significantly after switching off the regularizer: the optimization is not stuck. The point of convergence *does* change, just not in a way that improves test accuracy.

The fact that applying regularization only at the very beginning yields comparable results, suggests that regularization matters *not* because it alters the shape of the loss function at convergence, reducing convergence to spurious minimizers, but rather because it "directs" the initial phase of training towards regions with multiple extrema with similar generalization properties. Once the network enters such a region, removing regularization causes the solution to move to different extrema, with no appreciable change in test accuracy.

**Regularization delayed.** In this experiment, we switch on regularization starting at some epoch $t_0$, and continue training to convergence. We train the DNNs for 200 epochs, except when regularization is applied late (from epoch 150/175), where we allow the training to continue for an additional 50 epochs to ensure the network's convergence. Figure 1 (Center) displays the final accuracy as a function of the onset $t_0$, which shows that there is a "critical period" to perform regularization (around epoch 50), beyond which adding a regularizer yields no benefit.

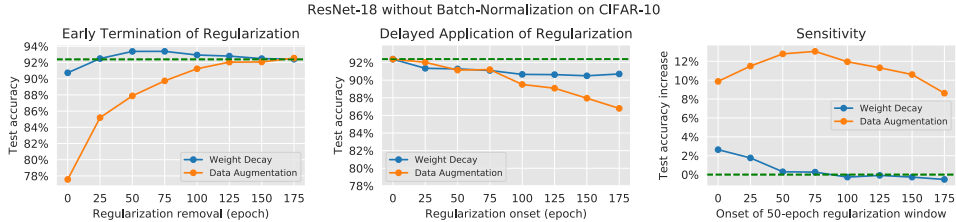

Figure 4: **Critical periods for regularization are independent of Batch-Normalization:** We repeat the same experiment as in Figure 1, but without Batch-Normalization. The results are largely compatible with previous experiments, suggesting that the effects are not caused by the interaction between batch normalization and regularization. **(Left)** Notice that, surprisingly, removal of weight decay right after the initial critical period actually improves generalization. **(Center)** Data augmentation in this setting shows a more marked dependency on timing. **(Right)** Unlike weight decay which mainly affects initial epochs, data augmentation is critical for the intermediate epochs.

Absence of regularization can be thought of as a form of learning *deficit*. The permanent effect of temporary deficits during the early phases of learning has been documented across different tasks and systems, both biological and artificial [1]. *Critical periods* thus appear to be fundamental phenomena, not just quirks of biology or the choice of the dataset, architecture, learning rate, or other hyperparameters in deep networks.

In Figure 1 (Top Center), we see that delaying WD by 50 epochs causes a 40% increase in test error, from 5% regularizing all along, to 7% with onset $t_0 = 50$ epochs. This is despite the two optimization problems sharing the same loss landscape at convergence. This reinforces the intuition that WD does not improve generalization by modifying the loss function, lest Figure 1 (Center) would show an increase in test accuracy after the onset of regularization.

Here, too, we see that the optimization is not stuck in a local minimum: Figure 2 (B) shows the weights changing even after late onset of regularization. Unlike the previous case, in the absence of regularization, the network enters prematurely into regions with multiple sub-optimal local extrema, seen in the flat part of the curve in Figure 1 (Center).

Note that the magnitude of critical period effects depends on the kind of regularization. Figure 1 (Center) shows that WD exhibits more significant critical period behavior than DA. At convergence, data augmentation is more effective than weight decay. In Figure 3 (Center), we observe critical periods for DNNs trained on CIFAR-100, suggesting that they are independent of the data distribution.

**Sliding Window Regularization.** In an effort to regress which phase of learning is most impacted by regularization, we compute the maximum sensitivity against a sliding window of 50 epochs during which WD and DA are applied (Figure 1 Right). The early epochs are the most sensitive, and regularizing for a short 50 epochs yields generalization that is almost as if we had regularized all along. This captures the *critical period for regularization*. Note that the shape of the sensitivity to critical periods depends on the type of regularization: Data augmentation has essentially the same effect throughout training, whereas weight decay impacts critically only the initial epochs. Similar to the previous experiments, we train the networks for 200 epochs except, when the window onsets late (epoch 125/150/175), where we train for 50 additional epochs after the termination of the regularization window which ensures that the network converges.

**Reshaping the loss landscape.** $L_2$ regularization is classically understood as trading classification loss against the norm of the parameters (weights), which is a simple proxy for model complexity. The effects of such a tradeoff on generalization are established in classical models such as linear regression or support-vector machines. However, DNNs need not trade classification accuracy for the $L_2$ norm of the weights, as evident from the fact that the training error can always reach zero regardless of the amount of $L_2$ regularization. Current explanations [11] are based on asymptotic convergence properties, that is, on the effect of regularization on the loss landscape and the minima to which the optimization converges. In fact, for learning algorithm that reduces to a convex problem, this is the only possible effect. However, Figure 1 shows that for DNNs, the critical role of regularization is to change the dynamics of the initial transient, which biases the model towards regions with good

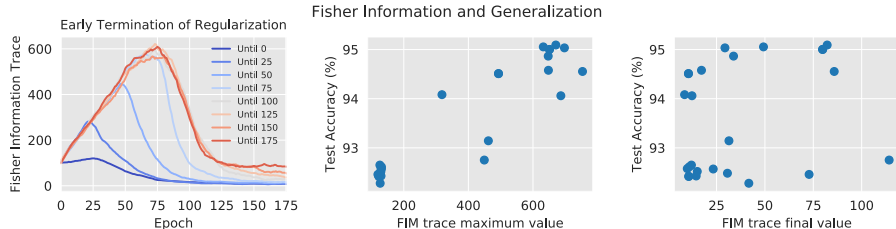

Figure 5: **Fisher Information and generalization: (Left)** Trace of the Fisher Information Matrix (FIM) as a function of the training epochs. Weight decay increases the peak of the FIM during the transient, with negligible effect on the final value (see left plot when regularization is terminated beyond 100 epochs). The FIM trace is proportional to the norm of the gradients of the cross-entropy loss. FIM trace plots for delayed application/sliding window can be found in the Appendix (Figure 7) **(Center) & (Right): Peak vs. final Fisher Information correlate differently with test accuracy:** Each point in the plot is a ResNet-18 trained on CIFAR-10 achieving 100% training accuracy. Surprisingly, the maximum value of the FIM trace correlates far better with generalization than its final value, which is instead related to the local curvature of the loss landscape ("flat minima"). The Pearson correlation coefficient for the peak FIM trace is **0.92** (p-value < 0.001) compared to **0.29** (p-value > 0.05) for the final FIM trace.

generalization. This can be seen in Figure 1 (Left), where despite halting regularization after 100 epochs, thus letting the model converge in the un-regularized loss landscape, the network achieves around 5% test error. Also in Figure 1 (Top Center), despite applying regularization after 50 epochs, thus converging in the regularized loss landscape, the DNN generalizes poorly (around 7% error). Thus, while there is reshaping of the loss landscape at convergence, this is not the mechanism by which deep networks achieve generalization. It is commonly believed that a smaller $L_2$ norm of the weights at convergence implies better generalization [37, 31]. Our experiments show no such causation: Slight changes of the training algorithm can yield solutions with larger norm that generalize better (Figure 2, (C) & Figure 1, Top right: onset epoch 0 vs 25/50).

**Effect of Batch-Normalization.** One would expect $L_2$ regularization to be ineffective when used in conjunction with Batch-Normalization (BN) [19], since BN makes the network's output invariant to changes in the norm of its weights. However, it has been observed that, in practice, WD improves generalization even, or especially, when used with BN. Several authors [38, 15, 35] have observed that WD increases the effective learning rate $\eta_{eff,t} = \eta_t/\|w_t\|_2^2$, where $\eta_t$ is the learning rate at epoch $t$ and $\|w_t\|_2^2$ is the squared-norm of weights at epoch $t$, by decreasing the weight norm, which increases the effective gradient noise, which promotes generalization [29, 20, 17]. However, in the sliding window experiment for $L_2$ regularization, we observe that networks with regularization applied around epoch 50, despite having smaller weight norm (Figure 2 (C), compare onset epoch 50 to onset epoch 0) and thus a higher effective learning rate, generalize poorly (Figure 1 Top Right: onset epoch 50 has a mean test accuracy increase of 0.24% compared to 1.92% for onset epoch 0). We interpret the latter (onset epoch 0) as having a higher effective learning rate during the critical period, while for the former (onset epoch 50) it was past its critical period. Thus, previous observations in the literature should be considered with more nuance: we contend that an increased effective learning rate induces generalization only insofar as it modifies the dynamics *during the critical period*, reinforcing the importance of studying *when* to regularize, in addition to how. In Figure 9 in the Appendix, we show that the initial effective learning rate correlates better with generalization (Pearson coefficient 0.96, p-value < 0.001) than the final effective learning rate (Pearson coefficient 0.85, p-value < 0.001).

We repeat the experiments in Figure 1 without Batch-Normalization (Figure 4). We observe a similar result, suggesting that the positive effect of weight decay during the transient cannot be due solely to the use of batch normalization and an increased effective learning rate.

**Weight decay, Fisher and flatness.** Generalization for DNNs is often correlated with the flatness of the minima to which the network converges during training [14, 24, 21, 4], where solutions corresponding to flatter minima seem to generalize better. In order to understand if the effect of regularization is to increase the flatness at convergence, we use the Fisher Information Matrix (FIM),

which is a semi-definite approximation of the Hessian of the loss function [27] and thus a measure of the curvature of the loss landscape. We recall that the Fisher Information Matrix is defined as:

$$F := \mathbb{E}_{x \sim \mathcal{D}'(x)} \mathbb{E}_{y \sim p_w(y|x)} [\nabla_w \log p_w(y|x) \nabla_w \log p_w(y|x)^T].$$

In Figure 5 (Left) we plot the trace of FIM against the final accuracy. Notice that, contrary to our expectations, weight decay increases the FIM norm, and hence curvature of the convergence point, but this still leads to better generalization. Moreover, the effect of weight decay on the curvature is more marked *during the transient* (Figure 5). This suggests that the peak curvature reached during the transient, rather than its final value, may correlate with the effectiveness of regularization. To test this hypothesis, we consider the DNNs trained in Figure 1 (Top) and plot the relationship between peak/final FIM value and test accuracy in Figure 5 (Center, Right): Indeed, while the peak value of the FIM strongly correlates with the final test performance (Pearson coefficient 0.92, p-value < 0.001), the final value of the FIM norm does not (Pearson 0.29, p-value > 0.05). We report plots of the Fisher Norm for delayed/sliding window application of WD in the Appendix (Figure 7).

The FIM was also used to study critical period for changes in the data distribution in [1], which however in their setting observe an anti-correlation between Fisher and generalization. Indeed, the relationship between the flatness of the convergence point and generalization established in the literature emerges as rather complex, and we may hypothesize a more complex bias-variance trade-off like a connection between the two, where either too low or too high curvature can be detrimental.

**Jacobian norm.** [38] relates the effect of regularization to the norm of the Gauss-Newton matrix, $G = \mathbb{E}[J_w^T J_w]$, where $J_w$ is the Jacobian of $f_w(x)$ w.r.t $w$, which in turn relates to norm of the networks input-output Jacobian. The Fisher Information Matrix is indeed related to the GN matrix (more precisely, it coincides with the generalized Gauss-Newton matrix, $G = \mathbb{E}[J_w^T H J_w]$, where $H$ is the Hessian of $\ell(y, f_w(x))$ w.r.t. $f_w(x)$). However, while the GN norm remains approximately constant during training, we found the changes of the Fisher-Norm during training (and in particular its peak) to be informative of the critical period for regularization, allowing for a more detailed analysis.

## 5  Discussion and Conclusions

We have tested the hypothesis that there exists a *"critical period"* for regularization in training deep neural networks. Unlike classical machine learning, where regularization trades off the training error in the loss being minimized, DNNs are not subject to this trade-off: One can train a model with sufficient capacity to zero training error regardless of the norm constraint imposed on the weights. Yet, weight decay works, even in the case where it seems it should not, for instance when the network is invariant to the scale of the weights, *e.g.*, in the presence of batch normalization. We believe the reason is that regularization affects the early epochs of training by biasing the solution towards regions that have good generalization properties. Once there, there are many local extrema to which the optimization can converge. Which to is unimportant: Turning the regularizer on or off changes the loss function, and the optimizer moves accordingly, but test error is unaffected, at least for the variety of architectures, training sets, and learning rates we tested.

We believe that there are universal phenomena at play, and what we observe is not the byproduct of accidental choices of training set, architecture, and hyperparameters: One can see the *absence* of regularization as a learning deficit, and it has been known for decades that deficits that interfere with the early phases of learning, or *critical periods*, have irreversible effects, from humans to songbirds and, as recently shown by [1], deep neural networks. Critical periods depend on the type of deficits, the task, the species or architecture. We have shown results for two datasets, two architectures, two learning rate schedules.

While our exploration is by no means exhaustive, it supports the point that considerably more effort should be devoted to the analysis of the transient dynamics of Deep Learning. To this date, most of the theoretical work in Deep Learning focuses on the asymptotics and the properties of the minimum at convergence.

Our hypothesis also stands when considering the interaction with other forms of generalized regularization, such as batch normalization, and explains why weight decay still works, even though batch

normalization makes the activations invariant to the norm of the weights, which challenges previous explanation of the mechanisms of action of weight decay.

We note that there is no trade-off between regularization and loss in DNNs, and the effects of regularization cannot (solely) be to change the shape of the loss landscape (WD), or to change the variety of gradient noise (DA) preventing the network from converging to some local minimizers, as without regularization in the end, everything works. The main effect of regularization ought to be on the transient dynamics before convergence.

At present, there is no viable theory on transient regularization. The empirical results we present should be a call to arms for theoreticians interested in understanding Deep Learning. A possible interpretation advanced by [1] is to interpret critical periods as the (irreversible) crossing of narrow bottlenecks in the loss landscape. Increasing the noise – either by increasing the effective learning rate (WD) or by adding variety to the samples (DA) – may help the network cross the right bottlenecks while avoiding those leading to irreversibly sub-optimal solutions. If this is the case, can better regularizers be designed for this task?

**Acknowledgments**

We would like to thank the anonymous reviewers for their feedback and suggestions. This work is supported by ARO W911NF-15-1-0564 and ONR N00014-19-1-2066.

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
