[Supplementary Material · NeurIPS_2019_supp.pdf]

# A    Details of the Experiments

We use the standard ResNet-18 architecture [12] in all the experiments, stated unless otherwise. We train all the networks using SGD (momentum = 0.9) with a batch-size of 128 for 200 epochs, except when regularization is applied late during the training, where we train for an extra 50 epochs, to ensure the convergence of the networks. For experiments with ResNet-18, we use an initial learning rate of 0.1, with learning rate decay factor of 0.97 per epoch and a weight decay coefficient of 0.0005. In the piece-wise constant learning rate experiment, we use an initial learning rate of 0.1 and decay it by a factor of 0.1 every 60 epochs. While in the constant learning experiment we fix it to 0.001. For the All-CNN [33] experiments, we use an initial learning rate of 0.05 with a weight decay coefficient of 0.001 and a learning rate decay of 0.97 per epoch. For All-CNN, we do not use dropout and instead we add Batch-Normalization to all layers.

## A.1    Path Plotting

We follow the method proposed by [24] to plot the training trajectories of the DNNs for varying duration of regularization (Figure 2 A). More precisely, we combine the weights of the network (stored at regular intervals during the training) for different duration of regularization into a single matrix $M$ and then project them on the first two principal components of $M$.

Figure 6: PCA-projection of the training paths for All-CNN on CIFAR-10.

# B    Additional Experiments

Figure 7: Trace of FIM as a function of training epochs for delayed and sliding window application of weight decay for ResNet-18 on CIFAR-10.

Figure 8: We repeat the experiments from Figure 1 and replace the exponential learning rate with a constant learning rate (lr = 0.001). However, those are inconclusive since the error achievable with a constant rate does not saturate performance on the datasets we tested them, with the architectures we used. Test error is almost twice than what was achieved with an exponential or piecewise constant learning rate. It is possible that careful tuning of the constant could achieve baseline performance and also display critical period phenomena.

Figure 9: **Effect of learning rate on generalization: (Top)** The effective learning rate $\eta_{eff,t} = \eta_t/\|w_t\|_2^2$, where $\eta_t$ is the learning rate at epoch $t$ and $\|w_t\|_2^2$ is the norm of weights at epoch $t$. [38, 15, 35], shown as a function of the training epoch for the experiments from Figure 1 (Top blue) / Figure 2. **(Left)** and **(Center)** The effective learning rate is higher throughout the training (which includes the critical periods) for models which generalize better (Figure 1 Top blue) **(Right)** When we apply weight decay in a sliding window, the effective learning rate for the dark blue curve (weight decay from 0-50) is higher during the critical period but lower afterwards, compared to light blue curves (weight decay from 25-75, 50-100). However, the dark blue curve yields a higher final accuracy (test accuracy increase of 1.92%) compared to light blue curves (test accuracy increase of 1.39%, 0.24% respectively), despite having a lower effective learning rate for the majority of the training. This suggests that having a higher effective learning rate during the critical periods is more conducive to good generalization when using weight decay. **(Bottom)** We test this further by plotting the relationship between the effective learning rate and final test accuracy. Mean initial effective learning rate is the average effective learning rate over the first 100 epoch, while mean final effective learning rate is the average computed over the final 100 epochs. The initial effective learning rate correlates better with generalization (Pearson correlation coefficient of **0.96** (p-value < 0.001)) compared to the final effective learning rate (Pearson correlation coefficient of **0.85** (p-value < 0.001)).

Figure 10: $L_2$ norm of the weights as a function of the training epoch. We observe similar results (compared to Figure 2) for different architectures (ResNet-18 and All-CNN) and different datasets (CIFAR-10 and CIFAR-100).

Figure 11: Plot of layer-wise Fisher Information for ResNet-18 trained on CIFAR-10. Unlike [1] we do not observe changes in the relative ordering of the21layers, which makes sense since unlike the experiments in [1], the underlying data distribution is not changing

# C  Extended Discussion

**You say that more analysis is needed, but you do not provide any. What are your insights beyond observing that adding regularization at different times changes the result?**

Our results show that the classical interpretation of regularization, that is of smoothing the loss landscape and removing spurious minima, is not explanatory. That means that decades of theory of regularization from Tikhonov onwards do not apply to Deep Learning, and we need to rethink generalization. We posit that regularization serves to influence the solution towards a basin of attraction, which is important, unlike the particular extremum within. Developing a new theory of regularization is well beyond the scope of a single conference paper, but we have planted a seed for new explorations of regularization in non-convex, high-dimensional, overparametrized systems.

**You say that your results apply to different networks, different datasets and different optimization, but you only try two of each. How can you claim your results conclusive?**

The fact that there *exists* a network, a dataset, and an optimization schedule for which the phenomenon (critical periods for regularization) is observed shows that the phenomenon exists. We do not expect it to exist for every network, every task, and every optimization. We do show, however, that it is not a quirk of a particular choice of model, hyperparameters etc. More investigation is needed to understand the sensitivity to critical regularization phenomena in different models, different datasets and different optimizations.

**You say that these results occur when using SGD. What about when using other optimization methods?**

Other adaptive optimization schemes, such as Adam, do not play well with $L_2$ regularization, so one cannot reach limit performance and therefore experiments to validate the presence of critical periods are inconclusive.

**You say that this is not because the solution is stuck in a local minimum, but even if the solution moves, it is possible that it is in a flat minimum and just moving there, so it is still possible that regularization reshapes the loss function?**

Indeed, but such reshaping does not alter the quality of the solution in any meaningful manner. What does alter the solution is decided in the early phases of learning.

**You claim that reduced weights does not cause regularization, but the plots in Fig. 5 are not convincing. How do they show that the value of the Fisher in the transient are causative?**

Please refer to the correlation coefficient (Lines 226-28), and Fig. 5 and Fig. 7 to judge how the final norm of the weights correlate with generalization, in comparison to how the peak Fisher information correlates.

**You say that interrupting regularization can improve generalization. Where can that be seen?**

See Figure 1 (All-CNN epoch 75 for WD) and Figure 4.

**Why 50 epochs windows in the sensitivity experiments?**

Because they are sufficient to exhaust most of the beneficial effects of regularization via weight decay.

**Dropout is a form of regularization, did you try that? Applying Dropout intermittently?**

Many more avenues of investigations have been opened by our observations. We are now in the process of exploring them.