[Reviews · NeurIPS 2019]

Reviewer 1



Strengths a. The paper is well-written and the authors are clear about their claims. b. The idea of critical periods during training with reference to regularization is interesting. If true, this would give a different way to think about generalization. c. The authors have performed a number of experiments with different configurations. Although, there are deficiencies mentioned below. Weaknesses Despite the above a I see the following problems. My primary concern is that while a number of results have been presented, they do not necessarily support the conclusions that have been drawn. a. I think that the claim that regularization biases the training toward a set of solutions (each of which is as good as any) is not sufficiently borne out by the experiments. The 'initial transient' is identified in the experiments as occurring at around the 100 epoch mark after which, it is claimed, regularization does not make a difference to generalization. The conclusion drawn from this is that the transient determines which set of solutions is selected, and once the set has been selected regularization has no further effect. I would suggest that another possible explanation for this is that the solution is already quite close to its optimum at the end of the identified transient. For the case of weight decay this would mean that the regularization term does not have a significant effect (since the weights are already small), although the L2 norm of the weights can still move around a bit if the minimum is flat. I think that to establish the claim it should be shown that at the end of the identified transient, the solution is still quite far from the optimum and simply comparing the norms might not be enough if the minimum is flat. Interestingly, this is recognized by the authors (lines 417-421) but not quite dealt with. b. Similarly for the case of delayed regularization I would suggest that the claim that after a period it has no effect is not sufficiently established. For example, one possible reason why delayed L2 regularization might have little effect is that weights have already become large before it is triggered. If so, then it might simply require more training time to lower the weights and reach convergence. That this is not ruled out is suggested by Figure 2 (B) where the weight norms are still on the decrease for delayed regularization. This seems to suggest that the models were not trained to convergence. c. I have another comment on the claim that it is a common belief that smaller L2 norms for weights lead to better generalization. I might be wrong but I don't think that this is commonly held. The cited paper [1] mentions this for linear problems y = Wx, but not as a general claim as far as I can tell. d. Datasets One weakness I find with the paper is that the experiments are only done for two very similar datasets. I think that more diversity in the datasets is required. The authors have anticipated this question (402-409), but I would counter by saying that one dataset and one network and one optimization procedure do not constitute a phenomenon. Reference: Wilson et al. The Marginal Value of Adaptive Gradient Methods in Machine Learning

Reviewer 2



This paper shows how regularization is a process with critical periods rather than a smoothing part of the loss function. While exploiting the same intuitions as a previous publication [1], it is clearly geared towards DNN practitioners, and hopefully theoreticians (rather than biologists) with a tutorial-like quality and great clarity. The connection (in related work) with stereo-view reconstruction is particularly well written and illuminating. While I have seen this issue being discussed, I have not come across a paper exploring so thoroughly the issue before, with a systematic exploration of all the cofounding factors: data distribution, learning rate, batch normalization. This is also the first time I have seen one dares to state, in the appendix to be cautious “That means that decades of theory of regularization from Tikhonov onwards do not apply to Deep Learning, and we need to rethink generalization.” The Fisher Information analysis seems similar to [1] but reaches sometimes opposite conclusions (correlation between Fisher and generalization). Figure 5 center, showing how the FIM trace max value correlates so well with test accuracy, is so counter-intuitive it could trigger new research if confirmed. One puzzling omission is any form of layer-wise analysis that could have better explained the strange results observed with FIM. The authors should explain why they are not showing such analysis (have they tried it?). Another decomposition, less trivial to try, would be sample-wise. Motivated by Fig.(2) in https://openreview.net/forum?id=SJfb5jCqKm, one could split examples along their confidence scores for a more detailed analysis.

Reviewer 3



The paper provides novel observations in terms of deep network regularization. The observations in the paper suggest that, for deep network training when to regularize the networks is critical; applying regularization at different phases of the training can yield different outcomes, and there is a critical period during the initial epochs of training which the regularization is more effective and after that, regularization may not have any benefit in terms of regularizing the networks to generalize well. These empirical results allow the authors to challenge the traditional view of regularization: altering the landscape of local extrema to which the model eventually converges. Basing on the new observations, the authors suggest that the “critical regularization period” of the regularization directs the training towards regions with multiple extrema with similar generalization properties. To support these claims, the authors conducted experiments with weight decay and simple data augmentation methods (i.e., random cropping and horizontal flipping) using two CNN architectures: ResNet-18 and All-CNN. The paper is very well written and easy to follow. The novel view that the timing is critical for regularizing the deep learning networks could have significant impact on the research field. The experiments conducted make sense to me. In particular, the observations provided in the paper could motivate researchers to focus on the transient rather than asymptotic behavior of the learning of the networks. On the other hand, experiments on more datasets, using other popular regualarizers, such as dropout and more sophisticated data augmentation methods (such as out-of-manifold regularizer Mixup[Mixup: Beyond empirical risk minimization, Zhang et. al.] and AdaMixup[Mixup as locally linear out-of-manifold regularization, Guo et. al.]), and with learning architectures beyond convolutional networks could significantly improve the paper. Also, some parts deserve more discussions such as the different behaviours of weight decay and data augmentation on Page6 in the section “Sliding Window Regularization”. Questions to the authors: 1. Is the critical regularization period correlated with the training loss? In this sense, it would be useful to plot the training loss curves of the models in the experimental section as well. 2. Is the critical regularization period data dependent? And how one could find this period? *********************after the rebuttal*********************** I would like to thank the authors for the rebuttal. The authors’ responses address most of my concerns. I like the additional experiments on two more data sets (i.e., SVHN and Imagenet) and on a recent Out-Of-Manifold data augmentation approach Mixup/AdaMixup. Also, the further clarification/discussion regarding the critical periods of regularization is also very helpful. I like this paper. *****************************************************************

[Author Response · NeurIPS 2019]



Figure 1: **(A)** Final test accuracy as a function of the weight decay termination/application epoch for All-CNN trained on SVHN dataset. Critical period for regularization occurs during the initial rapid decreasing phase of the training loss (red dotted line), which in this case is from epoch 0 to 75. **(B)** A ResNet-18 trained on ImageNet shows a critical period for data augmentation. **(C)** Plot of the $L_2$ norm of the gradients during training for ResNet-18 on CIFAR-10. **(D)** Plot of $L_2$ norm of the weights during training for ResNet-18 on CIFAR-10. The norm curves are almost parallel towards the end of the training, this suggests that further training will not reduce the weights significantly. **(E)** Plot of the test accuracy as a function of the regularization removal epoch for Mixup. **(F-H)** Layer-wise normalized Fisher Information of the weights as a function of training epochs for a ResNet-18 trained on CIFAR-10.

We thank the reviewers for their thoughtful analysis of the paper, and their suggestions, which we address below.

**R1:** Regularization may not necessarily bias the network toward a set equally good solutions, rather convergence to a
single (flat) optimum can explain the results.

This is a viable hypothesis that is not in conflict with our interpretation. After the end of the critical period, the
unregularized network may have converged to a (flat) local minimizer with sub-optimal generalization properties. Since
adding weight decay at this point does not change the performance of the network, we conclude that weight decay
does not (solely) help escaping bad local minimizers by reshaping the loss function. Rather, we claim that weight
decay and other regularizers help generalization by biasing the network toward particular minimizers during the initial
convergence phase. Note also that applying/removing regularization during the critical period (when the network is far
from convergence to a minimum, see Fig. 1C) still leads to partial improvement/deficits.

**R1:** Delayed application of WD may need more time to reach convergence.

In Fig. 1D we plot the norm of the weights for an additional 25 epochs to confirm that the norm of the weights stabilises
and would not improve further with additional training.

**R1, R3:** Testing on multiple datasets, architectures, and regularizers (*e.g.*, Mixup).

In the paper we verified our claims on a combination of several datasets, architectures, and regularizers, which is
sufficient to establish the claim that *"applying regularization at different epochs of training can yield different outcomes."*
Nonetheless, to assuage the reviewer's concern, we performed additional experiments on SVHN and on ImageNet (with
a correspondingly larger architecture), that further corroborate our result (Fig. 1A-B). As suggested by R3, we also try
the Mixup regularization on CIFAR-10 (Fig. 1E). In all cases we observe the same trends described in the paper.

**R2:** Lack of layer-wise analysis of Fisher. Another decomposition, less trivial to try, would be sample-wise.

See Fig. 1 (F-H) above for layer-wise analysis. Unlike [1] we do not observe changes in the relative ordering of the
layers, which makes sense since unlike the experiments in [1], the underlying data distribution is not changing. We will
work on sample-wise validation, thanks for the suggestion!

**R3:** Is the critical regularization period correlated with the training loss? It would be useful to plot the training loss
curves. Is the critical regularization period data dependent?

As also observed by [1], critical periods for regularization occur during the initial training epochs when the training
loss decreases rapidly, see Fig. 1 (A) and (E) (training loss is the red dotted line). Critical regularization periods do
depend on the dataset as different tasks (datasets) have different different learning dynamics and different responses to
the regularizers (some regularizer may be ineffective on some datasets), and thus different critical periods.

[Meta-Review · NeurIPS 2019]

The paper describes how regularization matters in different ways during different parts of the training processs, i.e., the timing is important for the regularization to be effective. Well written paper. Reviewers have several suggestions, which should be incorporated to the extent possible, but the ideas/results shoule be of interest to members of the community.